# Comparison of the Anti-Tumour Activity of the Somatostatin Receptor (SST) Antagonist [^177^Lu]Lu-Satoreotide Tetraxetan and the Agonist [^177^Lu]Lu-DOTA-TATE in Mice Bearing AR42J SST_2_-Positive Tumours

**DOI:** 10.3390/ph15091085

**Published:** 2022-08-30

**Authors:** Pascale Plas, Lorenzo Limana, Denis Carré, Amath Thiongane, Olivier Raguin, Rosalba Mansi, Florence Meyer-Losic, Stéphane Lezmi

**Affiliations:** 1IPSEN Innovation, 91940 Les Ulis, France; 2Oncodesign, 21000 Dijon, France; 3Division of Radiopharmaceutical Chemistry, Clinic of Radiology and Nuclear Medicine, University Hospital Basel, University of Basel, 4031 Basel, Switzerland

**Keywords:** [^177^Lu]Lu-satoreotide tetraxetan, [^177^Lu]Lu-DOTA-TATE, somatostatin receptor subtype 2, AR42J, peptide receptor radionuclide therapy

## Abstract

Limited experiments have compared the treatment effects of repetitive cycles of radiolabelled somatostatin (SST) analogues. In vitro and in vivo experiments were conducted in an AR42J cancer cell model, comparing the antagonist [^177^Lu]Lu-satoreotide tetraxetan with the agonist [^177^Lu]Lu-DOTA-TATE in terms of their binding properties, biodistribution, anti-tumour activity and toxicity. Histopathological and immunohistochemical examinations were performed at different timepoints. In the in vitro assays, [^177^Lu]Lu-satoreotide tetraxetan recognised twice as many SST_2_ binding sites as [^177^Lu]Lu-DOTA-TATE. In mice treated once a week for four consecutive weeks, [^177^Lu]Lu-satoreotide tetraxetan (15 MBq) revealed a significantly greater median time taken to reach a tumour volume of 850 mm^3^ (68 days) compared to [^177^Lu]Lu-DOTA-TATE at 15 MBq (43 days) or 30 MBq (48 days). This was associated with a higher tumour uptake, enhanced DNA damage and no or mild effects on body weight, haematological toxicity, or renal toxicity with [^177^Lu]Lu-satoreotide tetraxetan (15 MBq). At the end of the study, complete tumour senescence was noted in 20% of animals treated with [^177^Lu]Lu-satoreotide tetraxetan, in 13% of those treated with [^177^Lu]Lu-DOTA-TATE at 30 MBq, and in none of those treated with [^177^Lu]Lu-DOTA-TATE at 15 MBq. In conclusion, repeated administrations of [^177^Lu]Lu-satoreotide tetraxetan were able to potentiate peptide receptor radionuclide therapy with a higher tumour uptake, longer median survival, and enhanced DNA damage, with a favourable efficacy/safety profile compared to [^177^Lu]Lu-DOTA-TATE.

## 1. Introduction

Somatostatin analogues are an important treatment for well-differentiated, low-grade, gastroenteropancreatic neuroendocrine tumours (GEP-NETs) now. Peptide receptor radionuclide therapy (PRRT) targeting the somatostatin receptor subtype 2 (SST_2_) with the SST_2_ agonist [^177^Lu]Lu-DOTA-TATE, administered in four cycles of 7.4 GBq (200 mCi) per cycle, has become an established treatment modality for patients with advanced, well-differentiated, low-grade, SST_2_-positive GEP-NETs [1]. More recently, SST_2_ antagonists, especially [177Lu]Lu-satoreotide tetraxetan (also known as [^177^Lu]Lu-IPN01072, [^177^Lu]Lu-OPS201, or [^177^Lu]Lu-DOTA-JR11), have shown superior non-clinical and early clinical outcomes compared to the agonists and, as a result, research interest has shifted from radiolabelled SST_2_ agonists to antagonists, [2,3,4,5]. 

In the clinical setting, [^177^Lu]Lu-satoreotide tetraxetan has shown 1.1–2.6 times higher tumour uptake compared to [^177^Lu]Lu-DOTA-TATE [5]. In a phase I study in 20 patients with heavily pre-treated NETs, [^177^Lu]Lu-satoreotide tetraxetan therapy was associated with a median progression-free survival (mPFS) of 21 months and a disease control rate (DCR) of 85% [6], which is comparable to the DCR of 79% reported in a meta-analysis of 13 clinical studies in metastatic NETs treated with [^177^Lu]Lu-DOTA-TATE [7]. Compared with the SST_2_ agonists [^177^Lu]Lu-DOTA-TOC [4] and [^177^Lu]Lu-DOTA-TATE [3], in vitro and in vivo experiments showed that [^177^Lu]Lu-satoreotide tetraxetan had a 10-fold higher activity bound to human carcinoid cells transfected with SST_2_ [4], induced more DNA double-strand breaks (DSBs) [3], and was associated with a longer median survival [3,4] and a longer mean period of tumour stabilisation [3]. Similar differences between the antagonist [^177^Lu]Lu-DOTA-LM3 and the agonist [^177^Lu]Lu-DOTA-TOC were also observed [8].

Despite these promising results, to date, there has been no in vivo study comparing the treatment effects and the toxicity of four repetitive cycles of ^177^Lu-satoreotide tetraxetan and SST_2_ agonists such as [^177^Lu]Lu-DOTA-TATE, which is conventionally applied as four cycles in routine clinical practice. Accordingly, the aim of this present in vivo/ex vivo study was to compare the anti-tumour activity and the potential toxicity of [^177^Lu]Lu-satoreotide tetraxetan to that of [^177^Lu]Lu-DOTA-TATE, which were both administered for a total of four cycles in mice bearing AR42J xenografts–a rat pancreatic cancer model expressing endogenous SST_2_. 

## 2. Results

### 2.1. In Vitro

In the in vitro assay, saturation binding of both [^177^Lu]Lu-satoreotide tetraxetan and [^177^Lu]Lu-DOTA-TATE suited a single-site model, with [^177^Lu]Lu-satoreotide tetraxetan recognising almost twice as many SST_2_ binding sites as [^177^Lu]Lu-DOTA-TATE (mean ± SEM Bmax, 3210 ± 220 fmol/mg protein for ^177^Lu-satoreotide tetraxetan versus 1790 ± 600 fmol/mg protein for [^177^Lu]Lu-DOTA-TATE). Similarly, [^177^Lu]Lu-satoreotide tetraxetan was associated with a greater binding affinity than ^177^Lu-DOTA-TATE, as reflected by a lower mean ± SEM K_D_ value (0.088 ± 0.01 nM for ^177^Lu-satoreotide tetraxetan versus 0.48 ± 0.2 nM for [^177^Lu]Lu-DOTA-TATE).

### 2.2. In Vivo Biodistribution of ^177^Lu-Satoreotide Tetraxetan versus [^177^Lu]Lu-DOTA-TATE

The biodistribution results (expressed in both MBq/g and %ID/g) at 96 h following the fourth injection of the radiopharmaceuticals are summarised in Figure 1a and in Table 1. Compared to [^177^Lu]Lu-DOTA-TATE at either 15 or 30 MBq, significantly higher tumour, kidney, and femur radioactivity uptakes were observed with [^177^Lu]Lu-satoreotide tetraxetan at 15 MBq (Figure 1a). Indeed, the mean tumour radioactivity uptake was up to three times higher with [^177^Lu]Lu-satoreotide tetraxetan at 15 MBq (3.5 MBq/g) than with [^177^Lu]Lu-DOTA-TATE at 15 MBq (1.0 MBq/g; *p* = 0.0007) or at 30 MBq (2.2 MBq/g; *p* = 0.019). Similarly, in the kidneys, the mean radioactivity uptake was up to eight times higher with [^177^Lu]Lu-satoreotide tetraxetan at 15 MBq (0.51 MBq/g) than with [^177^Lu]Lu-DOTA-TATE at 15 MBq (0.066 MBq/g; *p* = 0.002) or at 30 MBq (0.20 MBq/g; *p* = 0.012). [^177^Lu]Lu-satoreotide tetraxetan at 15 MBq was also associated with a mean femur radioactivity uptake (0.014 MBq/g) two times higher than with [^177^Lu]Lu-DOTA-TATE at 15 MBq (0.006 MBq/g; *p* = 0.002) or at 30 MBq (0.008 MBq/g; *p* = 0.013). Similar differences were observed with the results expressed in %ID/g (Table 1). Regarding tumour-to-organ ratios, the mean tumour-to-kidney ratio was higher overall with [^177^Lu]Lu-DOTA-TATE (particularly at 30 MBq) than with [^177^Lu]Lu-satoreotide tetraxetan (Figure 1b). Overall, the tumour-to-femur ratio was comparable with [^177^Lu]Lu-satoreotide tetraxetan and [^177^Lu]Lu-DOTA-TATE (particularly at 30 MBq) (Figure 1b; Table 1).

### 2.3. In Vivo Anti-Tumour Activity of [^177^Lu]Lu-Satoreotide Tetraxetan versus [^177^Lu]Lu-DOTA-TATE

All treatments induced a significant tumour growth delay compared to the vehicle control group (all *p* < 0.0001) (Figure 2a and Appendix A; Table 2). Following the first treatment cycle with [^177^Lu]Lu-satoreotide tetraxetan, tumours showed a continuous decrease in volume and stabilisation for up to 50 days, which was significantly greater than with [^177^Lu]Lu-DOTA-TATE at 15 MBq (*p* < 0.0001) and 30 MBq (*p* < 0.0001) (Figure 2a). [^177^Lu]Lu-DOTA-TATE therapy had a dose-dependent effect on tumour volume, as treatment with 30 MBq of [^177^Lu]Lu-DOTA-TATE was associated with a greater decrease in mean tumour volume compared to 15 MBq of [^177^Lu]Lu-DOTA-TATE (*p* ≤ 0.02) (Figure 2a). The [^177^Lu]Lu-satoreotide tetraxetan-treated group showed a longer median time to reach the volume of 850 mm^3^ (68 days) compared to all other groups (Figure 2b, Table 2).

### 2.4. Histopathological and Immunohistochemical Analyses of Tumours at 96 Hours Post-Administration of Last Treatment

In all sampled tumours, SST_2_ expression, which was primarily localised to the cell membrane, was very strong (H-score of 300) independently of the administered treatments (vehicle, [^177^Lu]Lu-satoreotide tetraxetan, or [^177^Lu]Lu-DOTA-TATE at any dose) (Figure 3). 

Compared to controls (Figure 4a), histopathological examination of tumours at 96 h after the fourth injection of the test substances showed a complete loss of mitoses following treatment with [^177^Lu]Lu-satoreotide tetraxetan at 15 MBq and [^177^Lu]Lu-DOTA-TATE at 30 MBq (Figure 4b). There was an important decrease in mitotic counts following treatment with [^177^Lu]Lu-DOTA-TATE at 15 MBq, although evidence of mitotic activity remained (Figure 4c). The presence of degenerating cancer cells with a vacuolated cytoplasm and/or accumulation of granular eosinophilic material was also noted (Figure 4b, arrowheads). Sirius red staining revealed slightly higher fibrosis in the [^177^Lu]Lu-satoreotide tetraxetan-treated tumours (Figure 4e) compared to the vehicle control tumours and [^177^Lu]Lu-DOTA-TATE-treated tumours (Figure 4d,f). In line with the higher tumour radioactivity uptake of [^177^Lu]Lu-satoreotide tetraxetan at 15 MBq as compared to [^177^Lu]Lu-DOTA-TATE at 15 or 30 MBq, [^177^Lu]Lu-satoreotide tetraxetan produced more DSBs than [^177^Lu]Lu-DOTA-TATE treatments, as reflected by a higher mean pH2AX H-score (Figure 4g–i).

### 2.5. Histopathology and Immunohistochemistry of Tumours at the End of Study 

Histologic examination of tumours at the end of experiments revealed the presence of two distinct tumour phenotypes: a re-growing/relapsing phenotype (Figure 5a–c) and a senescent-like phenotype (Figure 5d–f). Both phenotypes can be observed in different lobules in the same tumour. 

In re-growing/relapsing tumours or tumour lobules, mitotic figures were commonly seen (Figure 4a, arrowheads), which was associated with a positive nuclear staining for Ki67 (Figure 4b). There was no evidence of fibrosis in the re-growing tumours or tumour lobules (data not shown). pH2AX expression was also evident in re-growing/relapsing tumours (Figure 5c). However, this was mainly due to a pH2AX H-score > 200 in two animals treated with [^177^Lu]Lu-satoreotide tetraxetan (Figure 5h), suggesting that [^177^Lu]Lu-satoreotide tetraxetan was able to induce persistent DNA damages in these two tumours. 

In senescent tumours or tumour lobules, there was no evidence of mitotic activity, and degenerating cancer cells with cytoplasmic vacuolation were apparent (Figure 5d, arrowhead). There was also a lack of Ki67 staining in cancer cells (Figure 5e) and a lack of pH2AX expression (Figure 5f). In addition, minimal (grade 1) to mild (grade 2) fibrosis was noted in the stroma of most senescent tumours (data not shown). Treatment with [^177^Lu]Lu-satoreotide tetraxetan was associated with a higher rate of complete tumour senescence (20%) compared to [^177^Lu]Lu-DOTA-TATE therapy at 15 MBq (0%) or at 30 MBq (13%). Similarly, there was a higher proportion of tumours presenting with senescent lobules in the ^177^Lu-satoreotide tetraxetan-treated group (70%) compared to [^177^Lu]Lu-DOTA-TATE at 15 MBq (27%) or at 30 MBq (40%) (Figure 5g; Appendix A). [^177^Lu]Lu-satoreotide tetraxetan showed a statistically significant higher rate of total (*p* < 0.001) and partial (i.e., tumour with both phenotypes of lobules: senescent and regrowing) (*p* < 0.01) tumour senescence compared to [^177^Lu]Lu-DOTA-TATE at 15 MBq. In comparison with [^177^Lu]Lu-DOTA-TATE at 30 MBq, a trend of less senescent lobules was observed, and more data would be needed to conclude.

At the end of the experiment, SST_2_ was still highly expressed in both regrowing tumor and tumor lobules as well as in senescent tumor lobules (Figure 6).

### 2.6. Toxicity Evaluation

In normal organs, SST_2_ was moderately expressed in the adrenal medulla (Figure 7a), the zona glomerulosa (not illustrated), the myenteric and submucosal plexuses (Figure 7b), as well as in epithelial crypt cells in the intestine (Figure 7b, arrowhead). Weak SST_2_ expression was also detected in the spleen in rare lymphoid cells (Figure 7c) and in the femoral bone marrow in osteoblasts (Figure 7d). SST_2_ was not detected in the kidneys (Figure 7f) nor in the tail (injection site, not illustrated). 

Interestingly, mean radioactivity uptakes (MBq/g of tissue) of [^177^Lu]Lu-satoreotide tetraxetan at 15 MBq and [^177^Lu]Lu-DOTA-TATE at both 15 and 30 MBq were overall well-correlated with SST_2_ expression levels in different organs, except in the kidneys (Appendix A).

Throughout the study, all treatments were well-tolerated, causing only a transient reduction in animal body weights that did not exceed 2% of that at the baseline. Haematological assessment at 96 h after the fourth injection of the test substances showed that, when compared to the vehicle control group, ^177^Lu-satoreotide tetraxetan at 15 MBq, [^177^Lu]Lu-DOTA-TATE at 15 MBq, and [^177^Lu]Lu-DOTA-TATE at 30 MBq induced biologically similar decreases in mean white blood cells mainly attributed to lymphocyte count changes, as well as similar increases in mean red blood cell counts (Appendix A). All these changes were considered of minimal toxicological significance. At the end of the study, minimal and biologically similar increases in platelet, white blood cell, and lymphocyte counts were noted in the three treated groups (Appendix A). 

Histopathological analyses at 96 h post administration of last treatment (Figure 8) revealed that similar treatment-related toxicity was observed in all three study groups, which was mainly characterised by a mildly to moderately increased myeloid/erythroid ratio in the bone marrow (Figure 8b), as well as a mildly to moderately decreased extramedullary haematopoiesis in the spleen (Appendix A). 

In the kidneys, the presence of basophilic tubules (likely corresponding to tubular regeneration) was minimal and was noted in all treatment groups (Appendix A). No evidence of treatment-related tissue damage was noted in the jejunum, colon, adrenal, or tail (injection site). Similarly, at the end of the study, histopathological examination of the bone marrow showed comparable mild increases in the myeloid/erythroid ratio across all study groups (Appendix A). The presence of rare foci composed of eosinophilic vacuolated cells (likely corresponding to Mott cells) was additionally noted in the bone marrow of one animal treated with [^177^Lu]Lu-DOTA-TATE at 30 MBq and one animal treated with [^177^Lu]Lu-satoreotide tetraxetan at 15 MBq (Appendix A; Figure 8c, arrowheads). There were no other treatment-related microscopic observations in any of the organs examined.

## 3. Discussion

Using a tumour model endogenously expressing high levels of SST_2_, the present study demonstrated for the first time that the radiolabelled SST antagonist [^177^Lu]Lu-satoreotide tetraxetan, administered as four fractions of 15 MBq given 1 week apart, suppressed tumour growth and induced tumour regression, prolonged median time taken to reach 850 mm^3^, and reduced the rate of tumour relapse to a greater degree than [^177^Lu]Lu-DOTA-TATE administered as four fractions of either 15 or 30 MBq 1 week apart, without inducing a higher toxicity. Compared with [^177^Lu]Lu-DOTA-TATE at 15 or 30 MBq, ^177^Lu-satoreotide tetraxetan therapy was also associated with an approximately three times higher tumour uptake as well as a higher rate of DNA DSB damage, with limited effects on body weight, haematological toxicity, or renal toxicity. The enhanced in vivo anti-tumour activity of [^177^Lu]Lu-satoreotide tetraxetan compared to [^177^Lu]Lu-DOTA-TATE observed in the present study can be partially explained by the ability of [^177^Lu]Lu-satoreotide tetraxetan to recognise almost twice as many SST_2_ binding sites as [^177^Lu]Lu-DOTA-TATE, as shown by our in vitro saturation binding assays using AR42J rat pancreatic tumour membranes (mean Bmax, 3210 fmol/mg protein for [^177^Lu]Lu-satoreotide tetraxetan versus 1790 fmol/mg protein for [^177^Lu]Lu-DOTA-TATE). These in vitro results are also consistent with previous in vitro receptor binding assays using human embryonic kidney cells transfected with SST_2_, in which [^177^Lu]Lu-satoreotide tetraxetan was able to recognize about four times more specific binding sites than [^177^Lu]Lu-DOTA-TATE assuming that the internalization of both products in AR42J cells had the same profile [9] i.e., complete internalization of the agonist; whereas, more than 50% of the antagonist remained at the cell membrane, which has not been demonstrated in this model. Although the binding affinity of [^177^Lu]Lu-DOTA-TATE was lower than that of the [^177^Lu]Lu-satoreotide tetraxetan on AR42J membranes, Mansi et al. have shown a similar affinity for both products in hSST2-HEK-293 membranes [9]. This difference could be explained by species dissimilarities between human and rat SST_2,_ as shown with another SST_2_ agonist Pasireotide [10]. In addition, the transfected status of HEK-293 cells with the human SSTR_2_ gene may not reflect a normal expression of the receptor at the cell membrane as in a native environment, which is not the case for the AR42J model endogenously expressing SST_2__._

In the present study, the median tumour growth delay, which was defined as the time taken to reach a tumour volume of at least 850 mm^3^, was 68 days in the [^177^Lu]Lu-satoreotide tetraxetan-treated group compared with 43 and 48 days with ^177^Lu-DOTA-TATE at 15 and 30 MBq, respectively. The superiority of ^177^Lu-satoreotide tetraxetan over [^177^Lu]Lu-DOTA-TATE and other SST agonists such as [^177^Lu]Lu-DOTA-TOC in inhibiting tumour growth was also noted in other in vivo therapy studies, which applied different animal models and treatment regimens [3,4]. 

DNA damage detection is one of the most important biological markers in the assessment of pharmacodynamic effects of tumour treatments [11]. The radionuclide lutetium-^177^ is known to emit β-particles that can induce different types of DNA damage, among which DSBs are the most genotoxic [3]. However, capturing the kinetics of ^177^Lu-PRRT-induced DNA damage is complex as it is influenced by various factors including the biodistribution of the therapeutic agent within cells and organs, the time course of radiation delivery, and tissue clearance kinetics [12]. In this study, [^177^Lu]Lu-satoreotide tetraxetan produced more DSBs than [^177^Lu]Lu-DOTA-TATE, as reflected by a higher mean pH2AX H-score following the administration of four PRRT cycles. Even if radiation most likely induced direct cellular damage (on the DNA and other cell structures) or indirect damages (e.g., production of reactive oxygen species) affecting cancer cell survival, the detection of pH2AX depends on repair mechanisms that prevail over DNA DSB formation. Indeed, as phosphorylation of H2AX is part of the cell cycle checkpoints leading to apoptosis or correcting genetic lesions, pH2AX foci are thus theoretically only detected in dividing cells [12,13]. Hence, in organs with no-to-low dividing cell potential, although not detected by pH2AX immunohistochemistry, radioactivity uptake most likely induces DNA damage that might lead to transitory or long-term cell function impairments, but also to an earlier cellular senescence [14,15]. Interestingly, two animals treated with [^177^Lu]Lu-satoreotide tetraxetan presented with higher levels of DNA damage in relapsing tumours, suggesting a marked alteration of cell cycle regulation mechanism and very high levels of cellular damage. 

Cellular senescence is a common outcome of various anti-cancer interventions [14]. In this study, compared with [^177^Lu]Lu-DOTA-TATE treatments, [^177^Lu]Lu-satoreotide tetraxetan therapy was associated with a greater rate of complete tumour senescence as well as partial tumour senescence. In addition, the absence of mitosis and lack of Ki67 immunostaining in senescent tumours is strong evidence supporting the senescent status of some tumours at the end of the study, several days after the last treatment administration. In our opinion, senescent cancer cells have lost their dividing capacities due to radiation therapy and they will most likely slowly degenerate. As observed in Figure 5d, cancer cell vacuolation and denegation was noted in all senescent tumors/tumor lobules in our study. Recently, it has been shown in a mouse model of pancreatic ductal adenocarcinoma that therapy-induced tumour cell senescence promotes vascular remodelling through induction of a pro-angiogenic senescence-associated secretory phenotype, leading to enhanced drug delivery and T-cell infiltration that can sensitize tumours to different therapies such as chemotherapy and immunotherapy [16]. Hence, the tumour senescence outcomes obtained in our study with ^177^Lu-satoreotide tetraxetan therapy are encouraging and may translate clinically into improved therapeutic responses [17].

The acute toxicity of radiopharmaceuticals can be attributed to the targeting of rapidly dividing cells (e.g., bone marrow and other lymphoid organs) [18]. In our study, the weekly intravenous administration of [^177^Lu]Lu-satoreotide tetraxetan at 15 MBq, [^177^Lu]Lu-DOTA-TATE at 15 MBq, or 30 MBq for 4 weeks induced overall similar haematological and histological changes in the bone marrow, spleen, and kidneys, and these observations were considered of minimal toxicological significance. Whole body irradiation (WBI) was needed to optimize cancer cell survival and homogeneous tumour development after subcutaneous implantation to prevent immune-dependent rejection. This procedure could have impacted the effects of both agents, leading to increased toxicity and changes in tumour responses to radiotherapies. Considering that all mice were exposed to the same body irradiation dose 8 days before the first treatments, the comparison of toxicity and efficacy of [177Lu]Lu-DOTA-TATE and [177Lu]Lu-satoreotide tetraxetan remains relevant in our opinion. Indeed, even if WBI toxicity cannot be excluded completely, the impact of this procedure on the evaluation of the toxicity of radiotherapeutics is likely minimal, although it cannot be discarded completely. Regarding the impact of WBI on tumour response to the treatments, anti-tumour effects of both radiotherapeutics were well-correlated to the radioactivity uptake in tumours, indicating that differences between the treatments are unlikely influenced by WBI.

Interestingly, organ uptakes measured at 96 h after the last treatment administration were well-correlated with SST_2_ expression levels, except in the kidneys (Appendix A). Indeed, despite notable radioactivity uptake in the kidneys, there was no evidence of SST_2_ expression in this tissue. Renal uptake is most likely due to a non-specific reabsorption of peptides in the proximal part of the nephron by megalin and cubilin receptors [19]. In humans, as an amino acid solution containing arginine and lysine is used to saturate these renal receptors, the renal uptake and potential renal toxicity of radiopharmaceuticals is limited, as noted in several clinical studies of [^177^Lu]Lu-satoreotide tetraxetan [5,6,20].The low toxicity induced by both agonist and antagonist products could be due to the high SST2 expression in AR42J cells and a potential sink effect in the tumour limiting the radioactivity uptake in other organs. Indeed, the %ID/g of tissue (Table 1) or MBq/g of tissue (Appendix A) were higher in tumours compared to other organs for both products likely reducing other organs’ radioactivity exposure. However, our results at 96h could be compared with the results formerly published in the H69 cancer model with a lower SST2 expression with [177Lu]Lu-satoreotide tetraxetan [3]. In the AR42J model, the %ID/g in tumours was twice as high, reflecting a high SST2 expression in AR42J cells compared to H69. However, the %ID/g in other tissues was not consistent with a sink effect as the radioactivity uptake was higher or equivalent in the adrenal glands and kidneys, respectively, in our AR42J model and lower in the spleen when compared to the H69 model [3]. Therefore, even if there is more radioactivity uptake in tumours, the level of SST2 expression does not seem to be clearly associated with a change in radioactivity uptake in other organs. Finally, depending on the tumour size, degree of tumour fibrosis and blood perfusion, level of SST2 expression, and the amount of drug used to treat a patient, a sink effect in the tumour remains possible and expected but difficult to estimate in practice.

Overall, our encouraging efficacy and safety findings support the application of repeated treatment cycles of [^177^Lu]Lu-satoreotide tetraxetan. Fractionated dose PRRT protocols have been found to inhibit tumour growth and lead to reductions in tumour burden with decreased toxicity, by delivering a sufficient cumulative radioactivity to the tumour while allowing non-target tissues to recover [21]. Supporting the current clinical practice of performing several spaced cycles of PRRT, administration of [^177^Lu]Lu-satoreotide tetraxetan, at a median cumulative radioactivity of 13.0 GBq over three cycles separated by an 8–12-week interval, had a manageable safety profile with a 12-month DCR of 90% in an ongoing phase I/II study (ClinicalTrials.gov: NCT02592707) in 40 patients with progressive SST_2_-positive NETs [20]. The results of the current study suggest that PRRT with [^177^Lu]Lu-satoreotide tetraxetan may also be performed in four cycles at a higher median cumulative radioactivity, divided over four instead of three cycles, with a lower radioactivity per cycle compared to the radioactivity level administered in the ongoing phase I/II study.

## 4. Materials and Methods

### 4.1. Radioligands

The peptide conjugate satoreotide tetraxetan was supplied by Octreopharm/Ipsen (Berlin, Germany), and DOTA-TATE by Bachem (Bubendorf, Switzerland). ^177^LuCl3 (non-carrier added) was obtained from ITM (Munich, Germany). The preparation of [^177^Lu]Lu-DOTA-TATE and [^177^Lu]Lu-satoreotide tetraxetan has been described elsewhere [22]. For both compounds, the radiochemical purities were above 90%, the radiochemical yields above 97%, and their molar activities were identical (around 50 MBq/nmol).

### 4.2. Cell Culture and Membrane Preparation

The SST_2_-expressing AR42J, acinar pancreatic tumour cell line derived from a transplantable tumour of a rat exocrine pancreas, was provided by ATCC (LGC Standards, Molsheim, France). Cells were cultured in RPMI 1640 medium (Dutscher, Bernolsheim, France), supplemented with 10% foetal calf serum (PAN-Biotech, Aidenbach, Germany), in a humidified atmosphere at 37 °C with 5% CO2. AR42J membranes were obtained by sonication in 50 mM Tris–HCl, pH 7.4, and centrifugation at 39,000× *g* for 10 min at 4 °C. The pellet was resuspended in the same buffer and centrifuged at 50,000 ×*g* for 10 min at 4 °C, and membranes in the resulting pellet were stored at −80 °C until further use. 

### 4.3. Saturation Binding Assays

Saturation binding assays were performed in vitro using AR42J cell membrane suspension (10 µg/well) incubated with increased concentrations (ranging between 0.075 and 10 nM) of the radioligand. Non-specific binding of [^177^Lu]Lu-satoreotide tetraxetan was determined using 1000-fold excess of the unlabelled SST_2_ antagonist OPS202, also known as NODAGA-JR11. Non-specific binding of [^177^Lu]Lu-DOTA-TATE was determined using the same excess of the natural somatostatin-14 or DOTA-TATE. Membrane-bound radio-peptide was counted in a γ-counter (COBRA 5003, Packard Instruments). The number of binding sites per cell was calculated and fitted to a one-site binding model in GraphPad Prism 8 to determine the dissociation constant (K_D_) and the number of binding sites (Bmax). 

### 4.4. Animals and Ethics 

Healthy female Swiss Nude-Foxn1nu mice, of 7 weeks old, were obtained from Charles River (Ecully, France). The animals were housed and cared for according to French and European regulations on animal experimentation. All procedures with animals were submitted to the Institutional Animal Care and Use Committee of Oncodesign (CNREEA Agreement N°91). Animal physical well-being was monitored on a daily basis, and animal weight was measured at least twice a week. Tumours were induced by subcutaneously injecting 1 × 107 AR42J cells in 200 μL of RPMI 1640 into the right flank of the mice. In order to improve cell engraftment, AR42J tumour cell implantation was performed 72 h after whole-body irradiation with a γ-source (2 Gy 60Co; BioMep, Bretenières, France). 

### 4.5. In Vivo PRRT Studies

PRRT with either ^177^Lu-satoreotide tetraxetan or [^177^Lu]Lu-DOTA-TATE started once the tumours reached a mean volume of 150–250 mm^3^. A total of 62 mice were randomised according to their individual tumour volume into 4 groups: vehicle control group (group 1; n = 13), 15 MBq [^177^Lu]Lu-DOTA-TATE-treated group (group 2; n = 18), 30 MBq ^177^Lu-DOTA-TATE-treated group (group 3; n = 18), and 15 MBq ^177^Lu-satoreotide tetraxetan-treated group (group 4; n = 13). 15 MBq and 30 MBq of [^177^Lu]Lu-DOTA-TATE corresponded to 0.48 µg and 0.96 µg or 0.29 and 0.58 nano moles of peptide per mice, respectively (i.e., 24 µg/kg and 48 µg/kg or 14.5 nmol/kg and 29 nmol/kg, respectively), 15 MBq ^177^Lu-satoreotide tetraxetan corresponded to 0.59 µg or 0.29 nano moles of peptide per mice (i.e., 29.5 µg/kg or 14.5 nmol/kg). All test substances, including the mixture of radiolabelling buffer and 0.9% NaCl solution administered in the vehicle group, were intravenously injected into the caudal vein once a week for 4 consecutive weeks. The treatment schedule was determined by the growth kinetics of the AR42J tumour model. The day of randomisation was considered as day 0. In order to evaluate the anti-tumour activity of [^177^Lu]Lu-satoreotide tetraxetan and [^177^Lu]Lu-DOTA-TATE, the length and width of the tumours were measured thrice a week with calipers to determine the tumour volumes. Tumour growth delay was estimated by calculating the median time taken to reach a tumour volume of 850 mm^3^. Animals were sacrificed once their tumour volume exceeded 1500 mm^3^. The study was stopped on day 83 because only 1 and 3 animals remained with non-progressing tumours in groups 3 and 4, respectively.

### 4.6. In Vivo Biodistribution Studies

Twelve mice (3 in each group) were sacrificed at 96 h after the fourth administration of the test substances to evaluate at the same time toxicity and haematological parameters, and radioactivity uptake in tumours and organs after 4 cycles. The remaining 50 mice (10 in both groups 1 and 4, and 15 in both groups 2 and 3) were either sacrificed for ethical reasons on the last day of the study, on day 83. At 96 h, following the last treatment administration, individual tumours and organs of interest were collected, blotted dry, weighted, fixed in formalin, and the amount of radioactivity present in each organ was quantified using a γ-counter. The radioactivity uptake in tumour and organs was determined and expressed as MBq/g of tissue as well as percentage injected dose/g of tissue (%ID/g). 

### 4.7. Haematology, Histopathology and Immunohistochemistry 

At each of the sacrifice timepoints, blood samples were collected in EDTA tubes and complete heamatological analyses were performed using a VetScan HM5 (Abaxis) to investigate white blood cell, red blood cell and platelet parameters.

Once fixed, tumours, spleen, bone marrow (femur), jejunum, colon, tail (injection site), and kidneys were processed to paraffin blocks and tissues slides produced for immunohistochemical analyses or stained with haematoxylin and eosin (H&E) or sirius red (collagen staining) for histopathological examination. 

Regular immunohistochemistry was performed using heat-induced antigen retrieval, as previously described [23,24]. Briefly, sections were incubated with primary antibodies–pH2AX (ab81299 from, Abcam, Paris, France) and SST_2_ (ab134152, Abcam, Paris, France) or with an unspecific isotypic control antibody to assess the specificity of the labelling. Afterwards, sections were incubated with a biotinylated secondary antibody and then an amplification system (streptavidin-biotin-peroxidase, ABC vector) was used before revelation with 0.02% diaminobenzidine. For the streptavidin-biotin-alkaline-phosphatase procedure, the same steps were followed, with a fast red solution used as chromogen. Counterstaining was done using aqueous haematoxylin. 

All the slides were evaluated by a board-certified veterinary pathologist (S.L.) using a light microscope (Olympus BX41). The pH2AX and SST_2_ signal intensities were scored using a semi-quantitative approach (H-score), by multiplying the signal intensity score (0, absent; 1, weak; 2, moderate; 3, strong) with the percentage of labelled cells. The H-score, which ranges from 0 to 300, gives the relative weight of staining in a given tumour sample. Regarding the histopathological examination, the number of mitoses was counted in the areas of tumours with the highest mitotic figures in 1 high-power field at the objective 40. The severity of fibrosis was evaluated on a classical toxicologic pathology scale ranging from 0 (absent) to 5 (severe). 

### 4.8. Statistics

Statistical analysis was performed using GraphPad Prism v8 software (La Jolla, CA, USA), as well as JMP version 14.2 (SAS Institute Inc., Cary, NC, USA), R version 3.5.1 (https://www.r-project.org/) (accessed on date 1 October 2020)., and SAS version 9.4 (SAS Institute Inc., Cary, NC, USA) for the tumour growth and time to reach 850 mm^3^ analyses. The time for a tumour to reach a volume of at least 850 mm^3^ and was estimated using the Kaplan-Meier method. For comparisons, one- or two-way ANOVA with Tukey’s or Dunnett’s multiple comparison was used when appropriate. Fisher’s exact test was also used to compare total and partial senescence in tumours between the 4 groups at the end of the study. Data are presented as mean, standard deviation, standard error of the mean (SEM), median, and range. Asterisks were used to specify significance level, where * denotes *p* ≤ 0.05, ** ≤0.01, and *** ≤0.001.

## 5. Conclusions

In this study, using the AR42J tumour model, we have evaluated the in vivo efficacy and safety of the SST_2_ antagonist [^177^Lu]Lu-satoreotide tetraxetan, administered for four cycles at a radioactivity of 15 MBq per cycle, compared with the agonist [^177^Lu]Lu-DOTA-TATE given for four cycles at either 15 or 30 MBq per cycle, i.e., a human equivalent dose of 3.7 or 7.4 GBq per cycle. Both products were well-tolerated. However, treatment with [^177^Lu]Lu-satoreotide tetraxetan resulted in a higher tumour uptake, enhanced DNA damage, a longer median time to reach 850 mm^3^ and, a more delayed tumour growth. These favourable in vivo efficacy/safety results compared to the SST_2_ agonist [^177^Lu]Lu-DOTA-TATE frequently used in clinical practice, suggest that the SST_2_ antagonist [^177^Lu]Lu-satoreotide tetraxetan, at a relatively low-median cumulative radioactivity, could be an interesting candidate for PRRT.

## Figures and Tables

**Figure 1 pharmaceuticals-15-01085-f001:**
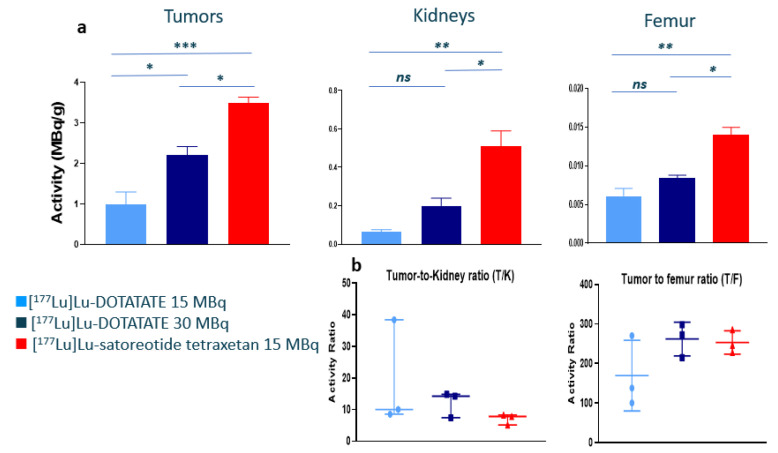
Biodistribution of [^177^Lu]Lu-DOTA-TATE at 15 and 30 MBq and [^177^Lu]Lu-satoreotide tetraxetan at 15 MBq at 96 h following the fourth injection of the radiopharmaceuticals. (**a**) Tumour, kidney, and femur radioactivity uptake expressed as mean ± standard error of the mean. * denotes *p* ≤ 0.05, ** ≤0.01, *** ≤0.001, and ns, non-significant. (**b**) Tumour-to-organ ratios expressed as median (range).

**Figure 2 pharmaceuticals-15-01085-f002:**
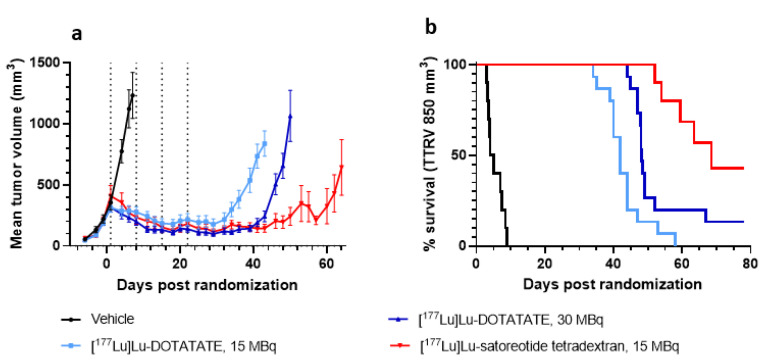
Anti-tumour effects of [^177^Lu]Lu-DOTA-TATE at 15 and 30 MBq and [^177^Lu]Lu-satoreotide tetraxetan at 15 MBq. (**a**) Tumour growth over time. Data are mean ± standard error of the mean. (**b**) Kaplan-Meier curves of Time to Reach Volume of 850 mm^3^ (TTRV850).

**Figure 3 pharmaceuticals-15-01085-f003:**
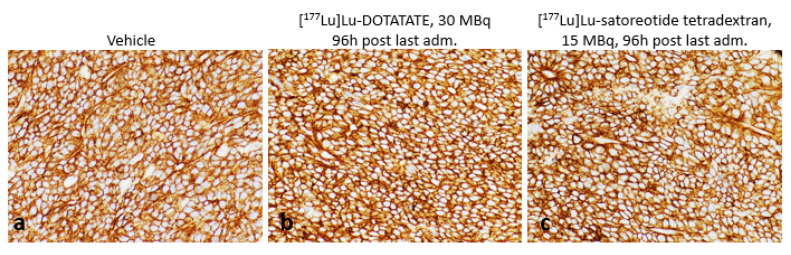
Somatostatin receptor subtype 2 (SST_2_) expression in tumours. The basal high SST_2_ expression (**a**) was not modified by the administered treatments either with [^177^Lu]Lu-DOTA-TATE (**b**) or [^177^Lu]Lu-satoreotide tetraxetan (**c**) 96h post last administration. All images were taken using an objective 40.

**Figure 4 pharmaceuticals-15-01085-f004:**
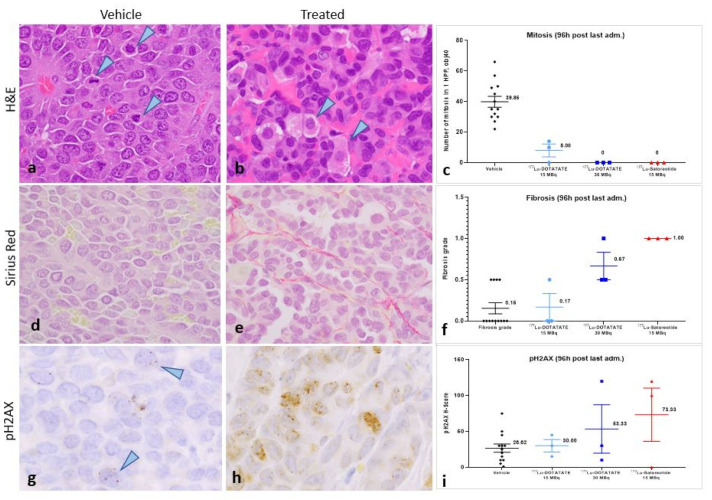
Histopathological and immunohistochemical analyses at 96 h post-administration of last treatment. (**a**,**b**) are representative images of haematoxylin and eosin (H&E) tumour staining. (**a**) The arrowheads point to mitoses in a control/untreated tumour. (**b**) In a [^177^Lu]Lu-satoreotide tetraxetan-treated tumour, mitoses were absent. The arrowheads point to degenerating cells. (**c**) Mean ± standard error of the mean (SEM) mitotic count (ranging from 22 to 66 at objective 40 in control tumours). (**d**,**e**) are representative images of sirius red staining. (**d**) In a control/untreated tumour, fibrosis was absent. (**e**) Fibrosis grade 1 (minimal) was noted in a [^177^Lu]Lu-satoreotide tetraxetan-treated tumour. (**f**) Mean ± SEM fibrosis score, ranging from 0 to 4. (**g**,**h**) are representative images of pH2AX tumour staining. (**g**) In a control/untreated tumour, there were low levels of staining in the nuclei of cancer cells (arrowheads) unlike in a [^177^Lu]Lu-satoreotide tetraxetan-treated tumour (**h**). (**i**) Mean ± SEM pH2AX H-score, ranging from 0 to 300. Figure (**a**,**b**,**d**,**e**) were taken at the objective 40, and figures (**g**,**h**) at the objective 100.

**Figure 5 pharmaceuticals-15-01085-f005:**
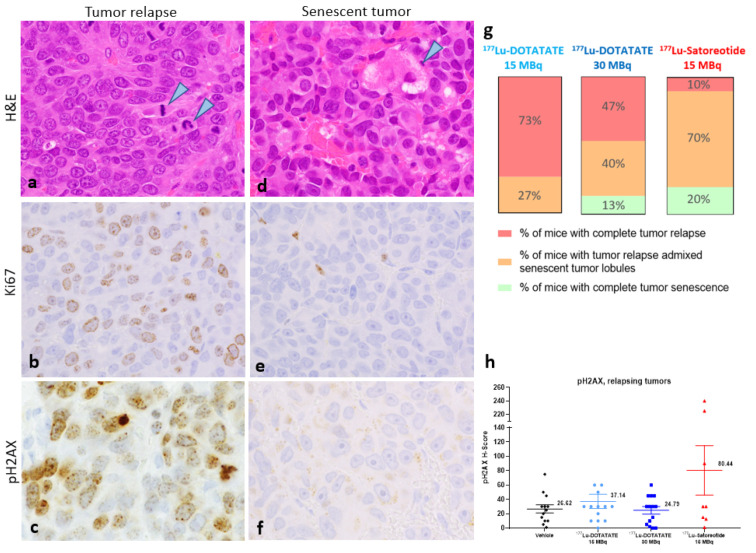
Histopathological and immunohistochemical analyses at the end of the study. Two distinct tumor phenotypes were noted: a re-growing/relapsing phenotype (**a**–**c**), and a senescent phenotype (**d**–**f**). In re-growing/relapsing tumours, mitotic figures were commonly seen (**a**, arrowheads) and associated with Ki67-positive staining (**b**). pH2AX expression was in some re-growing/relapsing tumours (**c**), which were treated with [^177^Lu]Lu-satoreotide tetraxetan (**h**, H-score > 200). In senescent tumours, there was no presence of mitoses of degenerating cancer (**d**, arrowhead) and a lack of Ki67 (**e**) and pH2AX (**f**) staining in cancer cells. All images were taken at the objective 100. (**g**) illustrates the proportion of mice with different tumour phenotypes in each treated group. (**h**) illustrates the mean ± standard error of the mean pH2AX H-score at the end of the study in re-growing/relapsing tumours.

**Figure 6 pharmaceuticals-15-01085-f006:**
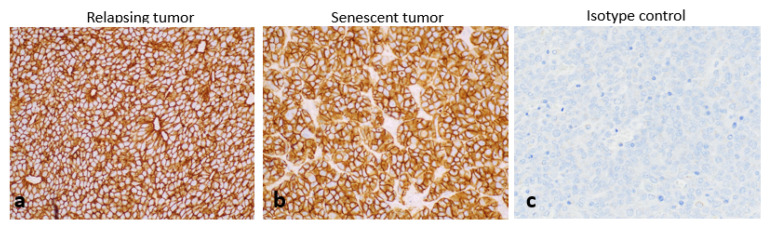
High expression of SST^2^ in relapsing tumours (**a**) and senescent tumours (**b**). The isotype control antibody confirmed the specificity of the staining (**c**). All images were taken using the objective 40.

**Figure 7 pharmaceuticals-15-01085-f007:**
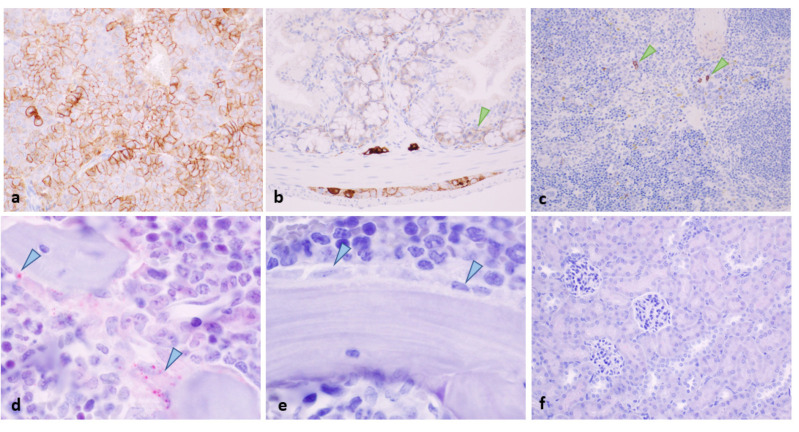
SST_2_ expression in the (**a**) adrenal medulla, (**b**) colon (including the enteric nervous system and epithelial crypt cells, arrowhead), (**c**) spleen (rare lymphoid cells, arrowheads), and (**d**) bone marrow osteoblasts (arrowheads). The isotype control antibody confirmed the specificity of the staining in the different tissues, as in the bone marrow (**e**). SSTR2 was not detected in the kidneys (**f**). Figures (**a**–**c**,**f**) were taken using the objective 40, and figures (**d**,**e**) were taken using the objective 100.

**Figure 8 pharmaceuticals-15-01085-f008:**
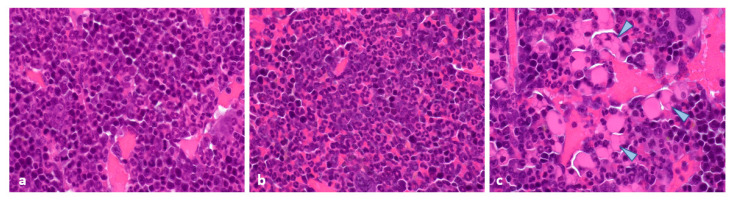
Histopathological analyses of the bone marrow at 96 h post administration of last treatment and at the end of study. Compared to a control bone marrow (**a**), similar treatment-related toxicity was observed in all three treated groups, which was mainly characterized by an increased myeloid/erythroid ratio (**b**) at 96 h after last administration. An increased myeloid/erythroid ratio was also noted at the end of the study in all study groups, with the presence of rare foci of eosinophilic vacuolated cells in a limited number of animals treated with either [^177^Lu]Lu-satoreotide tetraxetan at 15 MBq or [^177^Lu]Lu-DOTA-TATE at 30 MBq (**c**, arrowheads). Figures (**a**–**c**) were taken using the objective 100.

**Table 1 pharmaceuticals-15-01085-t001:** Biodistribution of ^177^Lu-DOTA-TATE at 15 and 30 MBq and ^177^Lu-satoreotide tetraxetan at 15 MBq and in AR42J pancreatic tumour-bearing mice.

Organs	[^177^Lu]Lu-DOTA-TATE 15 MBq	[^177^Lu]Lu-DOTA-TATE 30 MBq	[^177^Lu]Lu-Satoreotide Tetraxetan 15 MBq
Tumour	5.6 (1.7)	5.8 (0.58)	20 (0.80)
Kidneys	0.36 (0.067)	0.51 (0.11)	2.9 (0.44)
Adrenals	0.59 (0.082)	0.60 (0.048)	0.70 (0.34)
Femur	0.034 (0.0028)	0.022 (0.0011)	0.08 (0.0081)
Spleen	0.018 (0.0014)	0.015 (0.0015)	0.072 (0.0076)
Tail	0.055 (0.012)	0.042 (0.010)	0.082 (0.0080)

Organ uptake is expressed as mean (standard error of the mean) in units of percent injected dose per gram of tissue (%ID/g).

**Table 2 pharmaceuticals-15-01085-t002:** Median tumour growth delay (95% confidence interval).

Groups	Vehicle	[^177^Lu]Lu-DOTA-TATE 15 MBq	[^177^Lu]Lu-DOTA-TATE 30 MBq	[^177^Lu]Lu-Satoreotide Tetraxetan 15 MBq
**Time to reach tumour volume ≥850 mm^3^, days**	4.3 (2.5–7.7)	43 (40–45)	48 (47–49)	68 (51–not reached at end of study)
**Comparisons * between groups of time to reach tumour volume ≥850 mm^3^**				
**Vehicle**	Reference	*p* < 0.0001	*p* < 0.0001	*p* < 0.0001
**^177^Lu-DOTA-TATE 15 MBq**	-	Reference	*p* = 0.0036	*p* < 0.0001
**^177^Lu-DOTA-TATE 30 MBq**	-	-	Reference	*p* = 0.0044

* The adjusted *p*-values were determined by Chi-square test.

## Data Availability

Data is contained within the article and Appendix A.

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
