# Peer review of "Comparison of the Anti-Tumour Activity of the Somatostatin Receptor (SST) Antagonist [177Lu]Lu-Satoreotide Tetraxetan and the Agonist [177Lu]Lu-DOTA-TATE in Mice Bearing AR42J SST2-Positive Tumours"

_pharmaceuticals, 2022, doi:10.3390/ph15091085_

Round 1
Reviewer 1 Report
The authors report on the biological evaluation of the somatostatin receptor (SST) antagonist [177Lu]Lu-satoreotide tetraxetan in comparison with the agonist [177Lu]Lu-DOTATATE in mice bearing AR42J SST2-positive tumors. The manuscript is well written and shows an in vivo biodistribution and therapeutic study in a AR42J tumor-bearing mice model. The experimental section is very detailed from radiochemistry to in vivo tests.
However, this study is not new, in terms of investigating these two drugs, because there is already evidence of pilot clinical trials of these two drugs, which the authors mentioned in the water part of their manuscript. Nevertheless, this work complements existing studies by applying a strategy using 4 repetitive cycles of therapy in in vivo models.
But there are some minor points:
How significant is the tumor-to-kidney ratio in the case of [177Lu]Lu -DOTATATE 15 MBq, since the range of [177Lu]Lu -DOTATATE values at 15 MBq is relatively high?
Probably “an” should be used instead of “and” in the phrase «Somatostatin analogues have been and important».
Reviewer 2 Report
This paper seeks to compare the efficacy of agonist and antagonist ligands for delivery of somatostatin receptor 2-directed endoradiotherapy using clinical schedlues in a rodent pancreatic tumour model. Although the concept of superior antagonist efficacy has been well established in the literature, the novelty of this paper consists in the use of a clinically derived multiple dose schedule in a rodent model.
The authors compare a single dose of antagonist to two doses of antagonist, with tumor growth delay being the main output metric. However they usefully determine a useful range of histological measures on tumors and tissue ex-vivo after sacrifice to infer tissue and tumour responses.
As such it is I feel important enough to warrant publication in Pharmaceutics, and will be of great interest to the readership of the journal.
Major points:
Although the authors determine Bmax with each ligand in membrane assays, they do not characterize live cell uptake (membrane-bound vs internalized), and this is a significant omission.
The authors should also discuss that their model is a rodent one with a high levels of expression, contrasting this especially with Dalm et al (reference 3) and noting the potential 'tumour sink' effect on normal organ toxicity
In the materials and methods, it is stated 'In order to improve cell engraftment, AR42J tumour cell implantation was performed 72 hours after whole-body irradiation...'. The possible effects of whole body irradiation on subsequent therapy response should be discussed.
Staining for fibrosis should be included in the samples taken at the end of the experiment
Tumour activity at the end of the experiment, if available, should be correlated with DNA damage staining (if the hypothesis is that in some tumours, cell cycle arrest occurs whilst repair is undertaken, this should be confirmed by cell cycle analysis eg by FACS)
Individual tumour growth curves should be provided as supplementary data for the 'sensescent' antagonist-treated tumours
Minor points:
Grammar to be checked throughout
The abbreviation 'SST2' to be replaced by the more standard 'SSTR2'
Reviewer 3 Report
The reviewed publication is a continuation of the studies on the comparison of Lu-177 radiolabelled SST2 receptor antagonists and agonists. The work focuses on in-vivo studies, but in-vitro results are also presented.
My remarks:
In in-vitro studies, values of KD 0.088 nM for satoreotide tetraxetane and 0.48 nM for DOTATATE were obtained, while in Mansi et al Pharmaceuticals 2021; 14: 1265 the corresponding values are completely different 0.15 vs 0.09 nM. The authors should explain these discrepancies.
Also the authors many times discuss a much larger number of DSBs in aplication 177Lu satoreotide tetraxetane vs 177Lu-DOTATATE. However, the results of the DSB studies were not included in the publication and no references are available when results are discussed. This is an suprising result because beta radiation has a range much longer than the diameter of the cell and no difference should be observed. There is also no relevant information on whether the comparison was made, for the total radioactivity used in the experiments or for the cell-bound radioactivity. Because the bound radioactivity in using 177Lu-satoreotide tetraxetane is much greater than in the case of 177Lu-DOTATE it can be the source of more numbers DSB in 177Lu-satoreotide tetraxetane.
The in vivo studies have been done well and I have no comments on this part of the publication.
Round 2
Reviewer 2 Report
Many thanks to the authors for their point by point response. Some points would still benefit from additional clarification in my view, as indicated below:
i) cell studies. Although the point about beta range is correct, the issue here is the behaviour of murine SSTR2 receptor behaviour on agonist/antagonist binding. The comments do not address this, simply stating the AR42J cells are assumed to behave in the same was as HEK293 transfected with the human gene. It is important to know that in this system, the agonist is internalised and the antagonist membrane bound, as this is part of the mode of action. If this cannot be done, the assumption should be clearly stated in the text.
ii) tumour sink effect. The proposed text suggests that spleen uptake is 2 x as high, however I cannot see this is a significant difference given that the SEM is almost 50% of the mean. The kidney data is more important, however I still think it is important to note the high uptake in the tumors and the systemic effect this may have. The text should also state that tumor uptake is twice as high as that published in Dalm et al.
iii) effect of whole body irradiation. The question is whether WB irradiation affects the response to both therapies equally, and this should still be noted in the text as a variable in this set of experiments.
iv) fibrosis: points well argued and well taken
v) cell cycle: the point is well taken, but it would be useful to inlcude this brief synopsis in the discussion
Author Response
Please see the attachement
